# Performance Evaluation of Stator/Rotor-PM Flux-Switching Machines and Interior Rotor-PM Machine for Hybrid Electric Vehicles

**Wenfei Yu, Zhongze Wu \* and Wei Hua**

School of Electrical Engineering, Southeast University, Nanjing 210096, China; wenfeiyu@seu.edu.cn (W.Y.); huawei1978@seu.edu.cn (W.H.)
\* Correspondence: zzwu@seu.edu.cn

**Abstract:** A three-phase interior permanent magnet (IPM) machine with 18-stator-slots/12-rotor-poles and concentrated armature winding is commercially employed as a 10 kW integrated-starter-generator in a commercial hybrid electric vehicle. For comprehensive and fair evaluation, a pair of flux-switching permanent magnet (FSPM) brushless machines, namely one stator permanent magnet flux-switching (SPM-FS) machine, and one rotor permanent magnet flux-switching (RPM-FS) machine, are designed and compared under the same DC-link voltage and armature current density. Firstly, a SPM-FS machine is designed and compared with an IPM machine under the same torque requirement, and the performance indicates that they exhibit similar torque density; however, the former suffers from magnetic saturation and low utilization of permanent magnets (PMs). Thus, to eliminate significant stator iron saturation and improve the ratio of torque per PM mass, an RPM-machine is designed with the same overall volume of the IPM machine, where the PMs are moved from stator to rotor and a multi-objective optimization algorithm is applied in the machine optimization. Then, the electromagnetic performance of the three machines, considering end-effect, is compared, including air-gap flux density, torque ripple, overload capacity and flux-weakening ability. The predicted results indicate that the RPM-FS machine exhibits the best performance as a promising candidate for hybrid electric vehicles. Experimental results of both the IPM and SPM-FS machines are provided for validation.

**Keywords:** permanent magnet; brushless machine; stator-PM type; flux-switching; integrated-starter-generator; hybrid electric vehicle (HEV)

## 1. Introduction

An integrated-starter-generator (ISG) is a key component of hybrid electric vehicles (HEVs), which starts the engine as a motor and generates electric power as a generator according to operational conditions and system requirements [1–3]. At present, conventional permanent magnet (PM) brushless machines with PMs in the rotor are widely employed as ISG, e.g., an interior-PM (IPM) machine commercially used in a series of vehicles due to the high-power density, high efficiency and wide speed range [4,5]. As a key component of HEVs, research on ISGs has become a hot spot in the field of electrical machines. A dual air-gaps PM machine is investigated as an ISG in Ref. [1], where the torque characteristics and output power capability are verified using an analytical solution. In [6], a field-wound rotor synchronous machine and an IPM machine are compared under the same performance demands, and the results show that in the constant torque region, the IPM machine presents a higher efficiency, whereas in the high-speed constant power region, the two machines exhibit similar performances. In addition, a winding compression method for a V-type IPM machine used for ISG is presented to improve the winding thermal conductivity and power density with concentrated winding [7].

Despite the popularity of rotor-PM brushless machines, recently a type of flux-switching (FS) principle-based stator-PM brushless (SPM-FS) machine has attracted considerable attention, where both concentrated armature windings and PMs are located in stator and the rotor is made of salient iron laminations [8–10]. Obviously, the stator-PM topology favors good thermal dissipation and robust high-speed operation. However, due to the competition between PMs and armature for stator space, the SPM-FS machine suffers from significant stator teeth saturation due to the flux-concentrating effect, which limits the overload ability [9]. Hence, to alleviate the stator saturation and enhance torque capability, a novel FS machine with PMs in the rotor, named RPM-FS machine was proposed and analyzed [10]. The RPM-FS machine is also based on the flux-switching principle and consequently inherits the advantages of SPM-FS machines, while a significantly improved torque capability and overload capacity are also obtained by moving the PMs from stator to rotor [11].

It should be noted that, unlike a conventional spoke-type IPM machine, in an RPM-FS machine the modular rotor element is employed, and the magnetized directions of all magnets are the same instead of reversed. Previous research reveals that the RPM-FS machine can exhibit improved performance over SPM-FS and IPM machines used in Toyota-2004 [10]. However, the above results are based on the traction application where the rated and maximum power are 33 kW and 53 kW, respectively, and the rated and maximum speed are 1200 rpm and 6000 rpm, respectively. Therefore, in terms of the specifications and operation conditions for an ISG application, where a low DC voltage of 144 V and a large phase current of 150 A (RMS) are supplied, a pair of FS machines with PMs in the stator and rotor, respectively, namely, a SPM-FS machine and an RPM-FS machine, are designed and evaluated along with the IPM machine used in in a commercial HEV.

The arrangement of this paper is as follows. Firstly, the design specifications and topology features of an SPM-FS machine are introduced, and consequently electromagnetic performance comparison with the IPM-Honda machine is conducted in Section 2. Then, an RPM-FS machine with a multi-objective optimization algorithm is carried out by finite element method (FEM) in Section 3. Thereafter, comprehensive comparisons, including operation principle and electromagnetic performance of the pair of FSPM machines and the IPM machine, are discussed in Section 4. In Section 5, experimental validations on the prototyped machine are presented, and followed by conclusions highlighted in Section 6.

## 2. Design and Performance of the SPM-FS Machine

### 2.1. Design Specifications of ISG Application

Figure 1a shows the structure of a three-phase 18-stator-slot/12-rotor-pole (18 s/12p) IPM machine as an ISG. To evaluate the feasibility of FSPM machines for ISG application, Figure 1b shows a three-phase 12 s/10p SPM-FS machine. The stators of the two machines both employ modular elements in Figure 1c and concentrated tooth-wound-coils for armature windings. The topology differences are as follows.

- PM arrangement

The stator of the SPM-FS machine [Figure 1b] consists of U-shaped laminated segments, between which a piece of magnet is sandwiched and magnetized circumferentially in alternatively opposite directions. However, for the IPM machine, the PMs have a conventional inset configuration.

- Stator armature winding

The stators of the two machines are modularly manufactured with concentrate armature winding coils. However, since the PMs are sandwiched between modular U-core elements, the end-part length of the coils of the SPM-FS machine is longer than that of the IPM machine with the same stator diameter.

- Rotor

Compared with the IPM machine, the rotor of the SPM-FS machine is pressed by silicon steel sheets with good magnetic conductivity and mechanical strength, which is similar to that of switched reluctance machines.

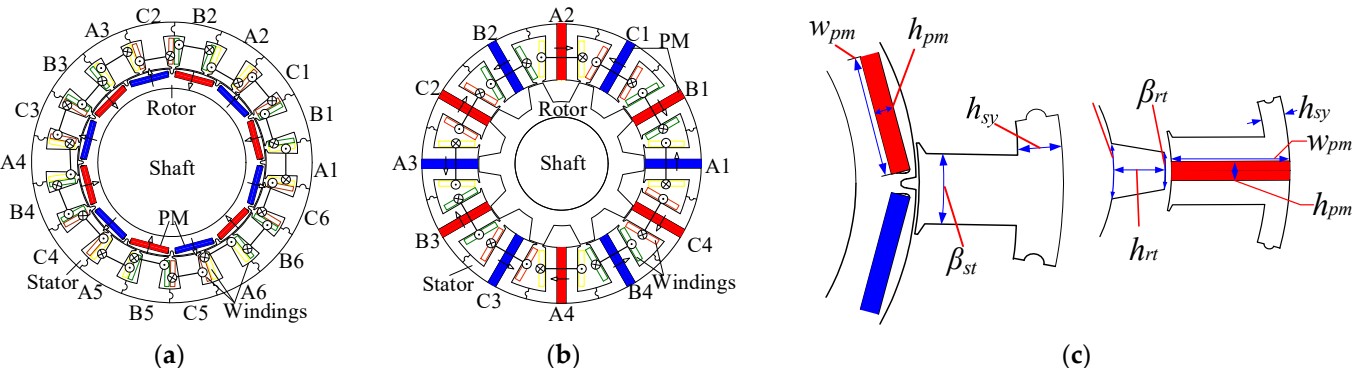

**Figure 1.** Topologies of the IPM machine and the SPM-FS machine with different PM locations. (**a**) IPM machine, (**b**) SPM-FS machine, (**c**) modular stator elements of the IPM machine and the SPM-FS machine.

### 2.2. Design of the SPM-FS Machine

To compare fairly, the key design specifications, material properties and machine dimensions of the SPM-FS machine are kept same as those of the IPM machine as illustrated in Table 1. $h_{pm}$ and $w_{pm}$ are variables that describe the size of the PMs in the IPM and SPM-FS machines, where $h_{pm}$ represents the thickness of the magnetization direction of the PM in both machines, and $w_{pm}$ represents the width of each pole of the PM in the machine cross-section. $g_0$ represents the single-side length of the air-gap, $\beta_{st}$ represents the stator tooth width, and $h_{sy}$ is the thickness of the stator yoke. In SPM-FS machines, $\beta_{rt}$ represents the width of the rotor tooth top, $\beta_{rty}$ represents the width of the rotor tooth bottom, and $h_{rt}$ represents the rotor tooth height. $k_{sio}$ is the stator split ratio, used to represent the proportion of the stator inner diameter $D_{si}$ to the stator outer diameter $D_{so}$. $k_{sio} = D_{si}/D_s$.

**Table 1.** Design parameters of the IPM and SPM-FS machines.

| Parameters | IPM | SPM-FS |
|:---|:---:|:---:|
| DC-link voltage, $U_{dc}$ (V) | 144 | 144 |
| Base speed, $n_N$ (rpm) | 1000 | 1000 |
| Rated torque, $T_N$ (Nm) | 95.5 | 95.5 |
| Max speed, $n_{max}$ (rpm) | 6000 | 6000 |
| PM remanence at 25 °C, $B_r$ (T) | 1.2 | 1.2 |
| PM coercive force at 25 °C, $H_c$ (A/m) | −868,292 | −868,292 |
| Stator/Rotor Iron material | 35WW300 | 35WW300 |
| Stator outer diameter, $D_{so}$ (mm) | 253 | 260 |
| Stack length, $L_a$ (mm) | 45.6 | 55 |
| Stator slot number, $P_s$ | 18 | 12 |
| Rotor pole number, $P_r$ | 12 | 10 |
| PM pole-pair number, $P_{PM}$ | 6 | 6 |
| Electromagnetic pole-pair number, $P_{fe}$ | 6 | 10 |
| Air-gap length, $g_0$ (mm) | 0.9 | 0.9 |
| Stator split ratio ($D_{si}/D_{so}$), $k_{sio}$ | 0.672 | 0.6 |
| Stator slot width, $\beta_{slot}$ (deg.) | 6.7 | 8 |
| Stator tooth width, $\beta_{st}$ (deg.) | 13.3 | 8 |
| Rotor tooth width, $\beta_{rt}$ (deg.) | / | 10.5 |
| Rotor yoke width, $\beta_{rty}$ (deg.) | / | 21 |
| Width of the magnet, $w_{pm}$ (mm) | 34.77 | 39 |
| Thickness of the magnet, $h_{pm}$ (mm) | 5.35 | 11.65 |
| Number of winding layers | 2 | 2 |
| Number of turns in series per phase | 60 | 48 |

**Table 1.** *Cont.*

| Parameters | IPM | SPM-FS |
|---|---|---|
| Number of strands | 20 | 17 |
| Number of parallel branches | 1 | 1 |
| Wire diameter, (mm) | 0.78 | 0.87 |
| Winding connection type | Y | Y |
| Magnet volume, $V_{pm}$ (mm$^3$) | 101,790 | 299,871 |
| Stator tooth width, $W_{st}$ (mm) | 20.3 | 12.4 |
| Stator yoke thickness, $T_{sy}$ (mm) | 12.4 | 9.5 |
| Turns number per coil, $N_c$ | 10 | 12 |

It can be found that the stator outer diameter ($D_{so}$ = 260 mm) and stack length ($L_a$ = 55 mm) of the SPM-FS machine are both larger than that of the IPM machine to satisfy the torque requirement ($T_N$ = 95.5 Nm@150Arms), resulting in a 27.4% larger volume than the IPM-machine due to the significant saturation in stator iron teeth as revealed in [9,12]. As the stator outer diameter is determined, the key stator dimensions in Figure 1c can be obtained initially as followed according to the design procedure [12],

$$\beta_{slot} = \beta_{st} = \beta_{pm} = 1/4\beta_{s\tau} \tag{1}$$

where $\beta_{slot}$ is stator slot width arc, $\beta_{st}$ is stator tooth width arc, $\beta_{pm}$ is magnet width arc, and $\beta_{s\tau}$ is the stator pole pitch arc, being equal to $D_{si}/P_s$ ($D_{si}$ is the stator inner diameter).

As for the rotor, it yields,

$$\beta_{ry} = 2\beta_{rt} = 2.625\beta_{st} \tag{2}$$

where $\beta_{rt}$ is the rotor tooth width arc and $\beta_{ry}$ is the rotor yoke width arc as shown in Figure 1c.

To maximize torque, the initial dimensions in (1) and (2) are changed slightly with optimization and the detailed parameters of the two machines are shown in Table 1.

### 2.3. Performance Comparison

In the following, the no-load and on-load performance of the IPM and SPM-FS machines are compared based on the finite element method (FEM), particularly on rated torque and overload capacity.

#### 2.3.1. Open-Circuit Flux Density Distribution

The open-circuit PM field distributions of the two machines are shown in Figure 2. For the IPM machine, due to a wider stator tooth width and yoke thickness (Table 1), the PM flux density in the stator iron is less than 1.2 T, and only a small portion of the stator teeth appears oversaturated. However, for the SPM-FS machine, significant saturation happens in the stator tooth due to the thinner tooth width, resulting in the limitation of torque output.

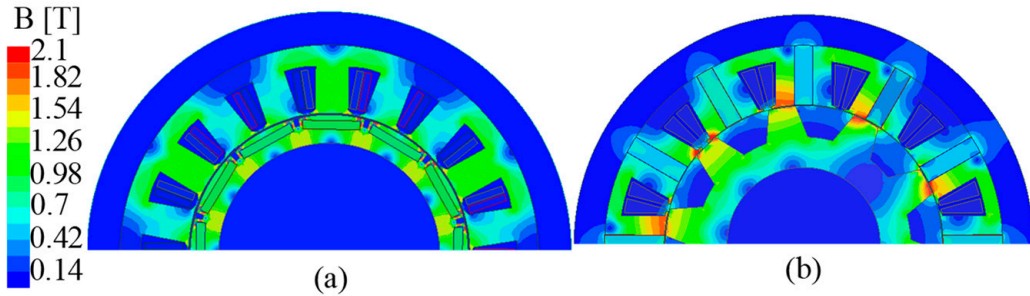

**Figure 2.** No-load flux density of the two machines. (**a**) IPM, (**b**) SPM-FS.

### 2.3.2. Open-Circuit Flux-Linkage, Back-EMF and Cogging Torque

Similarly, the PM flux-linkage and back electromotive force (back-EMF) per phase are compared in Figure 3. It can be seen that the phase PM flux-linkage waveforms of both machines are essentially sinusoidal. Based on the Fourier decomposition theory, the amplitude of the fundamental PM flux-linkage ($\Psi_{pm}$) of the IPM machine is 0.061 Wb and the total harmonic distortion (THD) is 1.15%, whereas for the SPM-FS machine, they are 0.053 Wb and 0.5%, respectively. Figure 3b shows the phase back-EMF of the two machines at the rated speed of 1000 rpm. It can be clearly seen that the back-EMF of the SPM-FS machine is more sinusoidal with a larger amplitude ($E_m$ = 55.45 V) and a lower THD of 2.34%. However, for the IPM machine $E_m$ = 38.55 V and the corresponding THD value is 7.04%. It can be seen that although the two machines employ concentrated windings, the back-EMF of the SPM-FS machine is significantly more sinusoidal.

Figure 3c compares the cogging torque waveforms of the two machines. Due to the different pole/slot combinations and rotor structures of the two machines, the period and peak to peak value of cogging torque are also different. The cogging torque period of the IPM machine and SPM-FS machine are 10° and 6° mechanical angle, respectively, and the peak-to-peak value are 9.28 Nm and 5.91 Nm, respectively. This means the SPM-FS machine exhibits a lower cogging torque and is favorable for reduction of torque ripple.

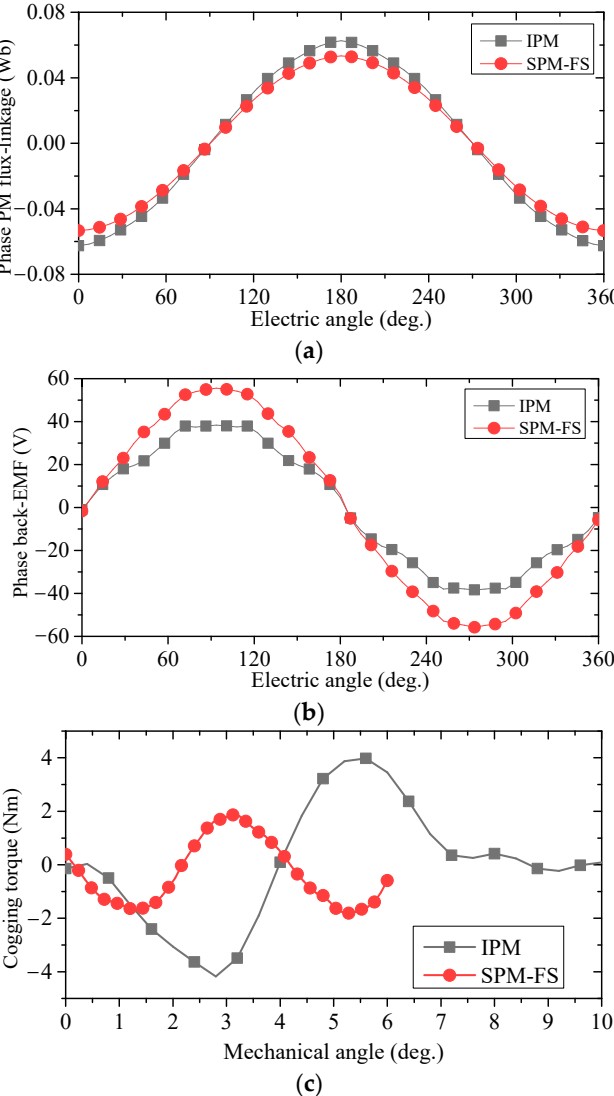

**Figure 3.** PM flux-linkage, back-EMF @1000 rpm and cogging torque waveforms of the two machines. (**a**) PM flux-linkage, (**b**) back-EMF. (**c**) cogging torque.

### 2.3.3. Static Rated Torque versus Current Angle

In addition to the no-load performance, the electromagnetic torque at the rated phase current of 150 A (rms) versus current angle ($\beta$) is shown in Figure 4. It can be seen that the optimal current angle to maximize torque are 20° and 10° for the IPM and SPM-FS machines, respectively, which agrees with the common sense notion that the IPM machines exhibit a considerable reluctance torque [13] due to the unequal $d$- and $q$-axis inductances ($L_d$ = 0.578 mH, $L_q$ = 0.448 mH @150Arms), whereas this is negligible for the SPM-FS machines [14] due to the almost equal $dq$-axes inductances ($L_d$ = 0.467 mH, $L_q$ = 0.431 mH @150Arms). On the other hand, it can be found that at the rated current, the torque of the SPM-FS machine is higher than that of the IPM machine by about 28%. However, it should be noted that the overall volume of the SPM-FS machine is larger than the IPM machine by around 27.4%. For the IPM machine, the effective volume of the machine including the winding end part is 3.3 L, and the volumes of the winding and PM are 0.34 L and 0.1 L, respectively. The volumes of the winding and PM account are 10.43% and 3.03% of the total volume of the machine, respectively. For the SPM-FS machine, the effective volume of the motor including the winding end is 3.98 L, and the volumes of the winding and PM are 0.26 L and 0.29 L, respectively. The volumes of the winding and PM account for 7.96% and 7.37% of the total volume of the machine, respectively. The PM usage of the SPM-FS machine is three times that of the IPM machine.

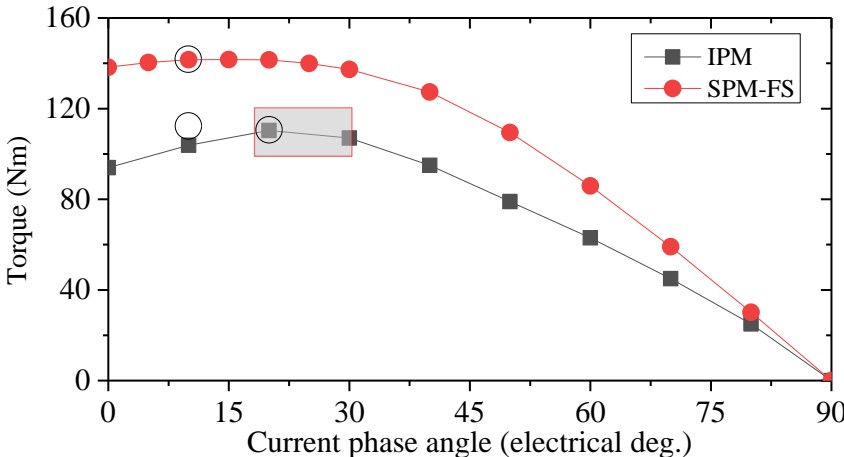

**Figure 4.** Effect of current angle on electromagnetic torque@ 150 Arms.

### 2.3.4. Torque-Current Capacity

To further evaluate the torque capacity of the two machines under the rated DC-link voltage of 144 V, the torque–current density ($T_e - J_{sa}$) curves are compared in Figure 5, where the maximum torque per ampere (MTPA) control is used based on the optimal current angle. It can be seen that the $T_e - J_{sa}$ curve of the IPM machine is always linear, even when the current is beyond the rated value of 150 Arms (corresponding to $J_{sa}$ = 20 A/mm$^2$) and the torque is 109.5 Nm. However, for the SPM-FS machine, the torque values are always higher than that of the IPM machine [15,16]. However, as the current increases beyond 105 A ($J_{sa}$ = 14 A/mm$^2$), the slope of torque significantly decreases due to the stronger stator iron saturation, as revealed in Ref. [17]. Another fact that should be emphasized is that the magnets consumed for the SPM-FS machine is almost three times that of the IPM machine as listed in Table 1, which indicates the SPM-FS machine is not a competitive candidate from the viewpoint of cost. Hence, in the next section a novel RPM-FS machine is introduced to address the above issues.

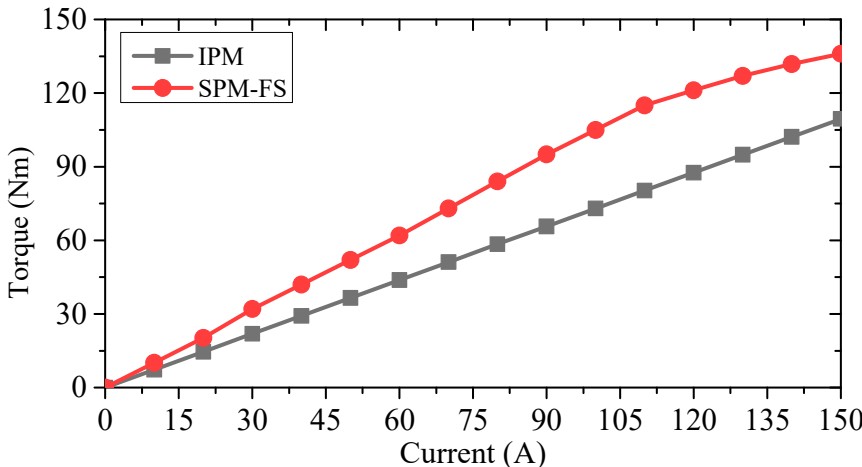

**Figure 5.** Torque versus current density of the IPM and SPM-FS machines.

## 3. Optimization Design of the RPM-FS Machine

To alleviate stator saturation and enhance torque capability, a novel RPM-FS machine is proposed, as shown in Figure 6 [10]. The outer diameter of the stator of the motor is 253 mm, and the stack thickness is 45.6 mm. The overall dimensions are consistent with the IPM machine. In Figure 6b, $h_{pm}$ and $w_{pm}$ are variables that describe the size of permanent magnets in RPM-FS machines, where $h_{pm}$ represents the thickness of the magnetization direction of the permanent magnet in both machines, and $w_{pm}$ represents the width of each pole of the permanent magnet in the machine cross-section. $\beta_{st}$ represents the stator tooth width, and $h_{sy}$ is the thickness of the stator yoke. $g_0$ represents the length of the air-gap. $\beta_{rtt}$ represents the width of the rotor tooth top, and $\beta_{rtb}$ represents the width of the rotor tooth bottom. Due to the moving of PMs from stator to rotor, a significantly improved torque capability and overload capacity has been witnessed in the applications in a Toyota Prius [11]. However, for ISG applications where a low DC voltage ($U_{dc}$ = 144 V) and a large current ($I_{rms}$ = 150 A) are required, the feasibility of RPM-FS machines should be evaluated further.

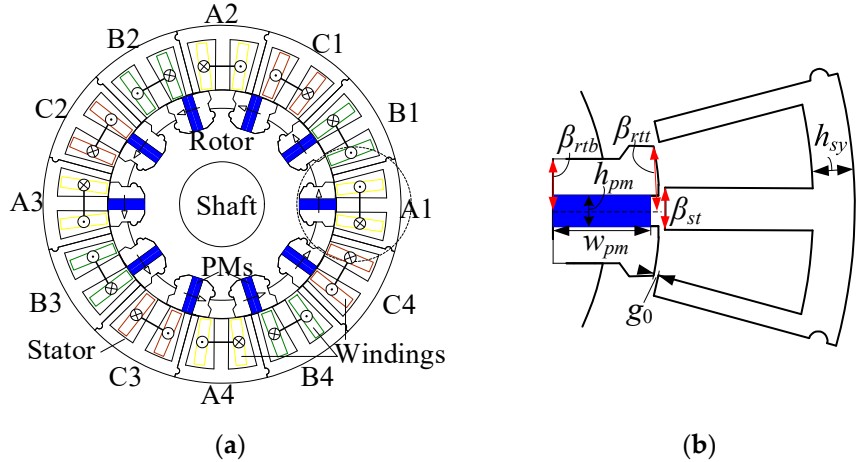

**Figure 6.** The 12/10 RPM-FS machine. (**a**) Cross-section, (**b**) modular element.

Recently, with the development of computer technology and optimization algorithms, design optimization of electrical machines has been gradually used [18,19]. Unlike the SPM-FS machine, the stator outer diameter and stack length of the RPM-FS machine are both kept the same as those of the IPM machine in addition to the other specifications. Since there are lots of design dimensions and evaluated electromagnetic performance, a multi-objective optimization is conducted based on FEM and genetic algorithm (GA).

Setting the optimization variables as $x_1, x_2, x_3 \ldots, x_n$, each performance index is a function of $f(x) = (x_1, x_2, x_3, \ldots, x_n)^T$,

$$\begin{cases} f_1(x) = f_1(x_1, x_2, x_3 \ldots x_n) \\ f_2(x) = f_2(x_1, x_2, x_3 \ldots x_n) \\ \qquad \cdots \\ f_n(x) = f_n(x_1, x_2, x_3 \ldots x_n) \end{cases} \tag{3}$$

Here, the performance indices include cogging torque, output torque, torque ripple, cost, material mass, efficiency and various losses. In addition, eight structural parameters are set as optimal variables and listed in Tables 2 and 3, while the detailed definitions are illustrated in Figure 6b. In order to facilitate the description of stator tooth width, stator yoke width, and rotor tooth top width, a radian measure is used for description. Since the RPM-FS machine does not have stator tooth shoes, it is initially assumed that the stator teeth and stator slot each account for half. Therefore, the stator tooth width is $k_{st} \times \pi/P_s$, where $k_{st}$ is the stator tooth width coefficient, the initial value is 1 and $P_s$ is the number of stator slots. The same thickness of the stator yoke is $k_{sy} \times \pi D_{si}/(4 \times P_s)$, where $k_{sy}$ is the thickness coefficient of the stator yoke, with an initial value of 2, and $D_{si}$ is the inner diameter of the stator. The width of the top of the rotor teeth is $k_{rtt} \times \pi/P_r$, where $k_{rtt}$ is the coefficient of the top width of the rotor teeth and $P_r$ is the number of rotor teeth. In the optimization design, the parameter variation ranges are listed in Table 4.

**Table 2.** Performance index of the RPM-FS machine.

| Performance Index | RPM-FS |
|---|---|
| Peak cogging torque, $T_{cog}$ (Nm) | $f(cog)$ |
| Rated torque, $T_N$ (Nm) | $f(torque)$ |
| Rated torque ripple, $T_{rip}$ (%) | $f(rip)$ |
| Material cost, ($) | $f(cost)$ |
| Material mass, $Q$ (kg) | $f(mass)$ |
| Efficiency, $\eta$ (%) | $f(eff)$ |
| Loss, (W) | $f(loss)$ |

**Table 3.** Optimal variable of the RPM-FS machine.

| Optimal Variables | RPM-FS |
|---|---|
| Stator split ratio, $(D_{si}/D_{so})$ | $k_{sio}$ |
| Rotor split ratio, $(R_{ri}/R_{ro})$ | $k_{rio}$ |
| Stator tooth width factor | $k_{st}$ |
| Stator yoke width factor | $k_{sy}$ |
| Rotor tooth top width factor | $k_{rtt}$ |
| Rotor tooth bottom width factor | $k_{rtb}$ |
| Magnet width factor | $k_{pmw}$ |
| Magnet length factor | $k_{pmh}$ |
| Stator tooth width arc, $(k_{st} \times \pi/P_s)$ | $\beta_{st}$ (deg.) |
| Stator yoke thickness, $(k_{sy} \times \pi D_{si}/(4 \times P_s))$ | $h_{sy}$ (mm) |
| Magnet width, $(k_{pmw} \times (R_{ro} - R_{ri}))$ | $w_{pm}$ (deg.) |
| Magnet length, $(D_{ri} \times \sin(k_{pmh} \times \pi/4 \times P_r))$ | $h_{pm}$ (mm) |
| Rotor tooth top width arc, $(k_{rtt} \times \pi/P_r)$ | $\beta_{rtt}$ (deg.) |

For the optimal design, the material cost yields

$$cost = 7.25 \times M_{Cu} + 29 \times M_{PM} + 1.45 \times (M_S + M_R) \tag{4}$$

where $M_{Cu}$, $M_{PM}$, $M_S$ and $M_R$ are mass of copper, PMs, stator and rotor silicon steel sheet, respectively. The coefficients 7.25, 29 and 1.45 in Equation (4) represent the prices of copper, permanent magnet and stator/rotor core during optimization (in $), respectively.

**Table 4.** Performance Index and Optimal Variable of the RPM-FS Machine.

| Parameters | Initial Value | Variation Range |
|---|---|---|
| $k_{sio}$ | 0.64 | 0.55~0.8 |
| $k_{st}$ | 0.96 | 0.8~1.2 |
| $k_{sy}$ | 1.9 | 1.8~2.4 |
| $k_{rio}$ | 0.68 | 0.6~0.7 |
| $k_{rrb}$ | 0.69 | 0.7~0.85 |
| $k_{rrt}$ | 0.7 | 0.7~0.85 |
| $k_{pmw}$ | 0.95 | 0.8~0.98 |
| $k_{pmh}$ | 0.855 | 0.7~1.2 |

In the optimization processes, the selected optimization objectives are efficiency, average torque, cogging torque, torque ripple and material cost. The constraints are phase current, DC-link voltage and PM mass. The optimization objectives and boundaries are listed in Table 5. The effects of design parameter variations on output torque, torque ripple, efficiency and cost of the RPM-FS machine are shown in Figure 7a,b. From Figure 7a, it can be seen that many cases can satisfy the conditions if only torque, torque ripple and efficiency are restricted. Figure 7b shows the case when the cost factor is added. Constraints such as performance, heat dissipation and cost need to be considered when determining the final solution. When selecting the RPM-FS optimization scheme, the optimal comprehensive performance scheme is ultimately selected from the perspectives of rated output torque greater than 125 Nm, efficiency greater than 92%, cost less than USD 45 and minimum torque ripple.

**Table 5.** The optimization objectives and boundaries.

| Indexes | Objective | Boundaries |
|---|---|---|
| Efficiency (%) | 0.64 | 0.55~0.8 |
| Torque (Nm) | 0.96 | 0.8~1.2 |
| Cogging torque (Nm) | 1.9 | 1.8~2.4 |
| Torque ripple (Nm) | 0.68 | 0.6~0.7 |
| Material cost ($) | 0.69 | 0.7~0.85 |
| Phase current (A) | 0.7 | 0.7~0.85 |
| Current density (A/mm$^2$) | 0.95 | 0.8~0.98 |
| Mass PM (kg) | 0.855 | 0.7~1.2 |

By further selection, the initial and optimal topologies are shown in Figure 8, and the design parameters are compared in Tables 6 and 7. It can be seen that the optimized design has a larger split ratio, smaller stator slot area, shorter stator teeth and increased widths of stator tooth and yoke to alleviate the saturation. Compared with the original one, the cogging torque, torque ripple and cost of the optimal model have decreased significantly, by 50.7%, 23.6% and 14.5%, respectively, whereas the rated torque and efficiency have increased by 38.2% and 1.65%, respectively.

**Table 6.** Comparison of the original and optimal RPM-FS Machines.

| Design Parameter | Initial Design | Optimal Design |
|---|---|---|
| $k_{sio}$ | 0.64 | 0.7464 |
| $k_{st}$ | 0.96 | 1.1813 |
| $k_{sy}$ | 1.9 | 2.3143 |
| $k_{rio}$ | 0.68 | 0.6133 |
| $k_{rrb}$ | 0.69 | 0.7045 |
| $k_{rrt}$ | 0.7 | 0.7206 |
| $k_{pmw}$ | 0.95 | 0.8897 |
| $k_{pmh}$ | 0.855 | 0.7223 |

**Table 7.** Comparison of the original and optimal RPM-FS Machines.

| Performance Index | Initial Design | Optimal Design |
|---|---|---|
| Cogging torque, $T_{cog}$ (Nm) | 5.38 | 2.65 |
| Rated efficiency, $\eta$ (%) | 90.5 | 92.1 |
| Rated torque, $T_N$ (Nm) | 93.4 | 129.1 |
| Rated torque ripple, $T_{rip}$ (%) | 8.9 | 6.8 |
| Cost, ($) | 48.44 | 41.42 |

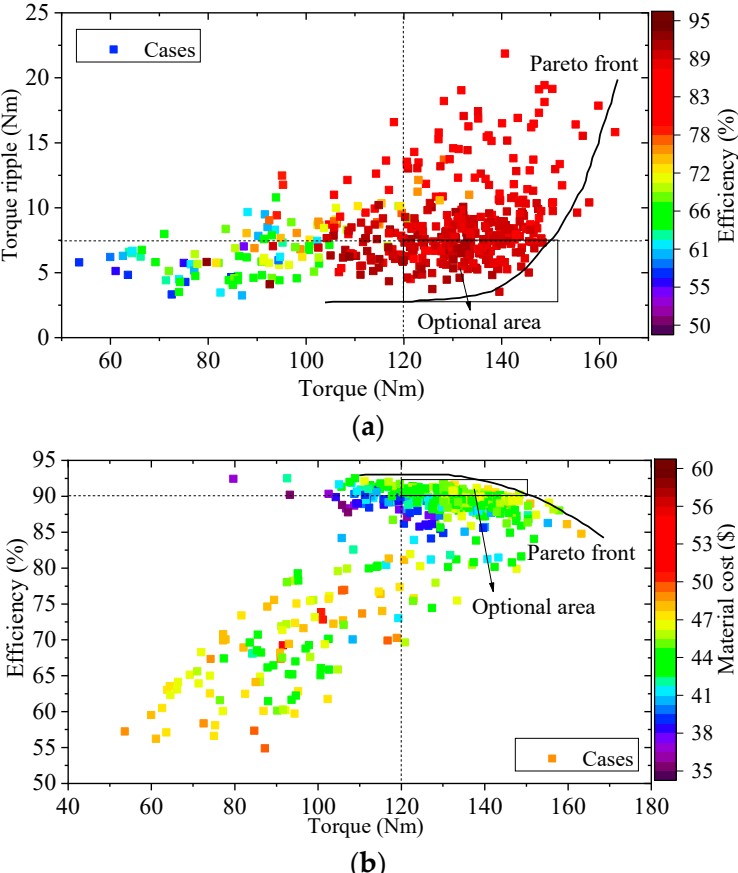

**Figure 7.** Optimization results of RPM-FS machine. (**a**) Considering torque, torque ripple, and efficiency, (**b**) considering torque, efficiency and material cost.

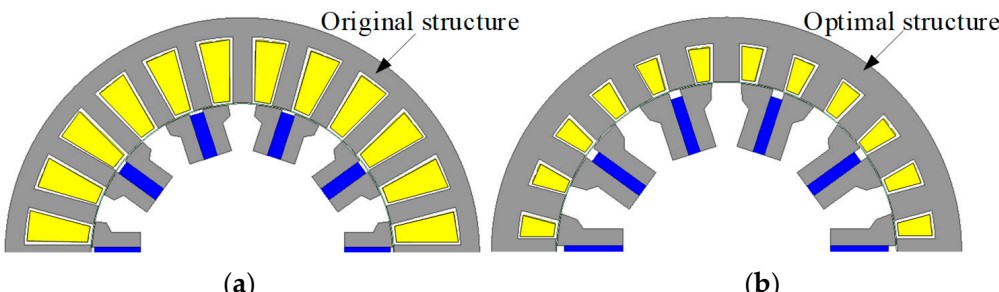

**Figure 8.** The original and optimal RPM-FS machines: (**a**) original, (**b**) optimal.

## 4. Performance Comparison of the Three Machines

In this section, a comprehensive comparison between the IPM, the SPM-FS machine and the optimized RPM-FS machines will be conducted based on the results above.

### 4.1. PM Flux Field Distribution and Air-Gap Flux Density

Figure 9a shows the open-circuit PM flux distributions of the three machines. To illustrate more clearly, the radial and tangential components of air-gap flux density are shown in Figure 9b,d. Based on the Fourier analysis, the radial and tangential components of air-gap flux density harmonic distributions are shown in Figure 9c,e. For the SPM-FS machine, the dominant harmonic components are the 6th and 18th, produced by the primitive PM magnetomotive force (MMF) (PM-MMF) ($nP_{PM}$, $n = 1$ and 3), whereas the other harmonics with 4, 8, 16 and 28 pole-pairs ($|nP_{PM} \pm kP_r|$, ($n = 1, k = 1$, and $n = 3, k = 1$) are generated since the PM-MMF is modulated by the salient rotor teeth in the air-gap field. Similarly, for the RPM-FS machine, the dominant harmonics produced by the PM-MMF are only the 10th component ($nP_{PM}$, $n = 1$ and 5); meanwhile, if the modulation of salient stator teeth to rotor PM-MMF is taken into consideration, the harmonics of the 14th and 34th components ($|nP_{PM} \pm kP_s|$, $n = 1, k = 1$) are generated. However, for the IPM machine, the dominant harmonics produced by the PM-MMF are only the 6th component ($nP_{PM}$, $n = 1$) and multiples of the sixth component.

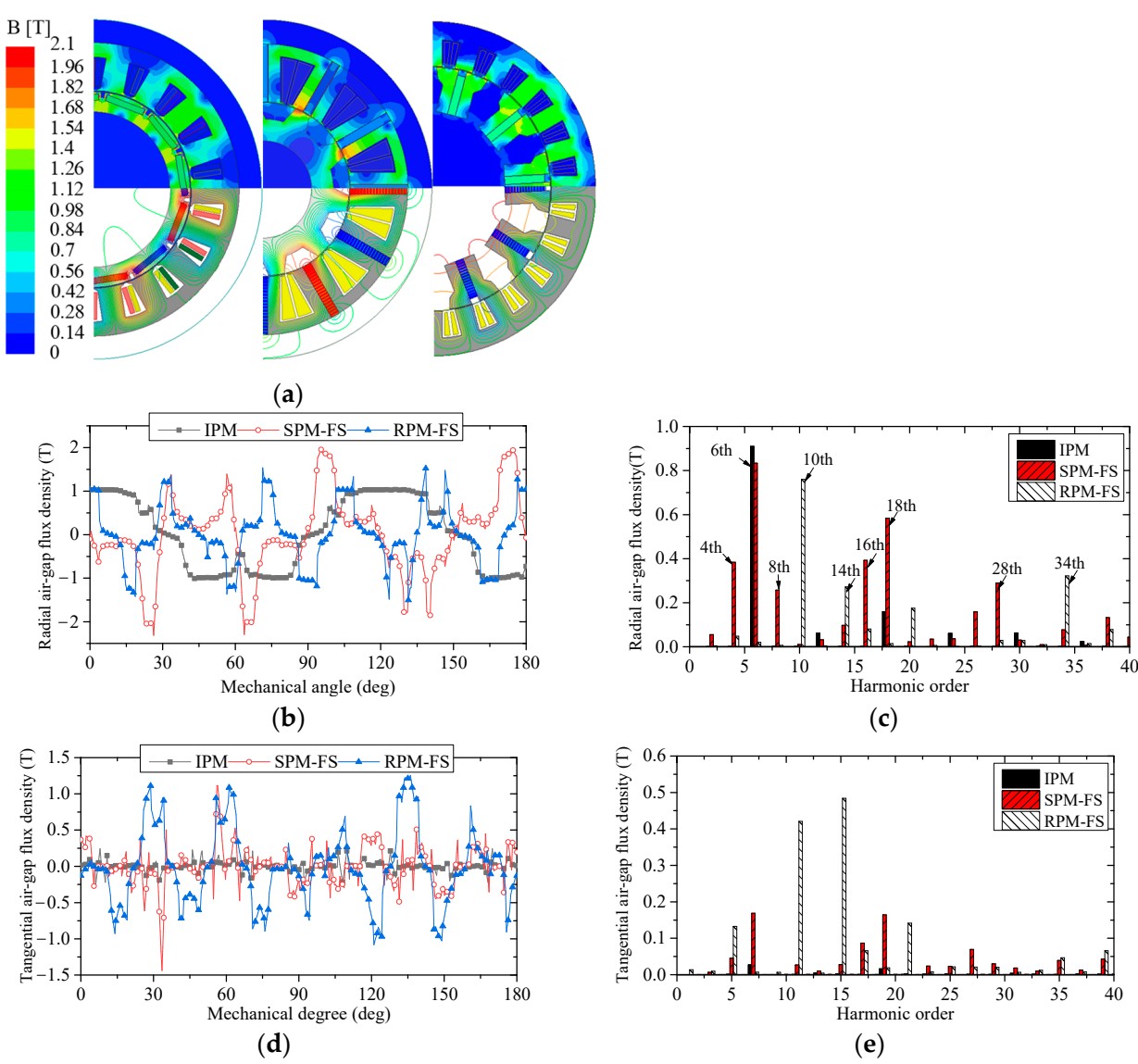

**Figure 9.** Open-circuit performance of the three machines. (**a**) PM field distribution, (**b**) radial air-gap PM flux density, (**c**) radial air-gap PM flux density harmonic distributions, (**d**) tangential air-gap PM flux density, (**e**) tangential air-gap PM flux density harmonic distributions.

*4.2. Cogging Torque*

For a PM machine with stator teeth number of $P_s$ and rotor pole number of $P_r$, the period of cogging torque $T_{cog}$ yields [20],

$$T_{cog} = \frac{360°}{LCM(P_s, \ P_r)} \tag{5}$$

where $LCM(P_s, P_r)$ is lowest common multiple of $P_s$ and $P_r$.

According to Equation (5), the period of the IPM, SPM-FS and RPM-FS machines are 10, 6 and 3 mechanical degrees, respectively, which agrees with the FEM prediction in Figure 10. Moreover, since the peak cogging torque is inversely proportional to $LCM(P_r, P_s)$, the peak–peak value of the IPM, SPM-FS and RPM-FS machines is, respectively 9.28 Nm, 5.91 Nm and 2.65 Nm, indicating that the two flux-switching machines exhibit lower cogging torque.

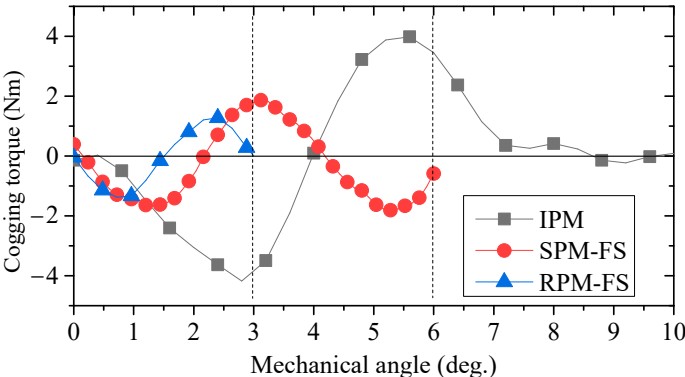

**Figure 10.** Predicted cogging torque waveforms of the three machines by FEM.

*4.3. End-Effect*

End-effect results from the axial flux leakage around the end-part of the stator/rotor iron in PM machines, leading to the decrease in no-load back-EMF amplitude and output torque [21]. Due to the specific application of ISGs, the ratios of stack length to stator outer diameter ($L_a/D_{so}$) of the three machines are all around 1/5, and hence the 3D end-effect cannot be neglected.

When calculating the air-gap flux density using 2D FEM, it is not possible to obtain the axial variation of the air-gap flux density at a certain position on the circumference. Instead, it is directly treated as a constant value. This equivalent method cannot take into account the attenuation effect of end effects on the air-gap flux density, as shown by the black line in Figure 11. To predict the end-effect on electromagnetic performance quantitatively, 3D-FEM models of the three machines are established. By taking the change of radial air-gap flux density at a certain position of the circumference along the axial direction, the red line in Figure 11 can be obtained. Both the 2D- and 3D-FEM based open-circuit air-gap flux density distributions in axial directions are shown in Figure 11. $S_{2D}$ and $S_{3D}$ represent the area of the envelope of the air-gap flux density along the axial direction of the machine based on 2D FEM and 3D FEM, respectively. The ratio of the area contained in the calculation of 2D and 3D flux densities is defined as the end effect. It can be seen that the amplitude of the air-gap flux density ($B_{gap}$) is almost unchanged along the axis in the iron region (half of stack length), while it tends to decrease around the end-part region of the iron. As shown in Figure 11, by comparing the surrounded areas composed of 3D- and 2D-FEM flux density waveforms, namely $S_{2D}$ and ($S_{2D} + S_{3D}$), respectively, the end-part factor can be calculated as 0.88, 0.89 and 0.87 for the IPM, SPM-FS and RPM-FS machines, respectively, which means during the initially design stage the influence of end-effect on performance should be considered to be approximately 10% quantitatively.

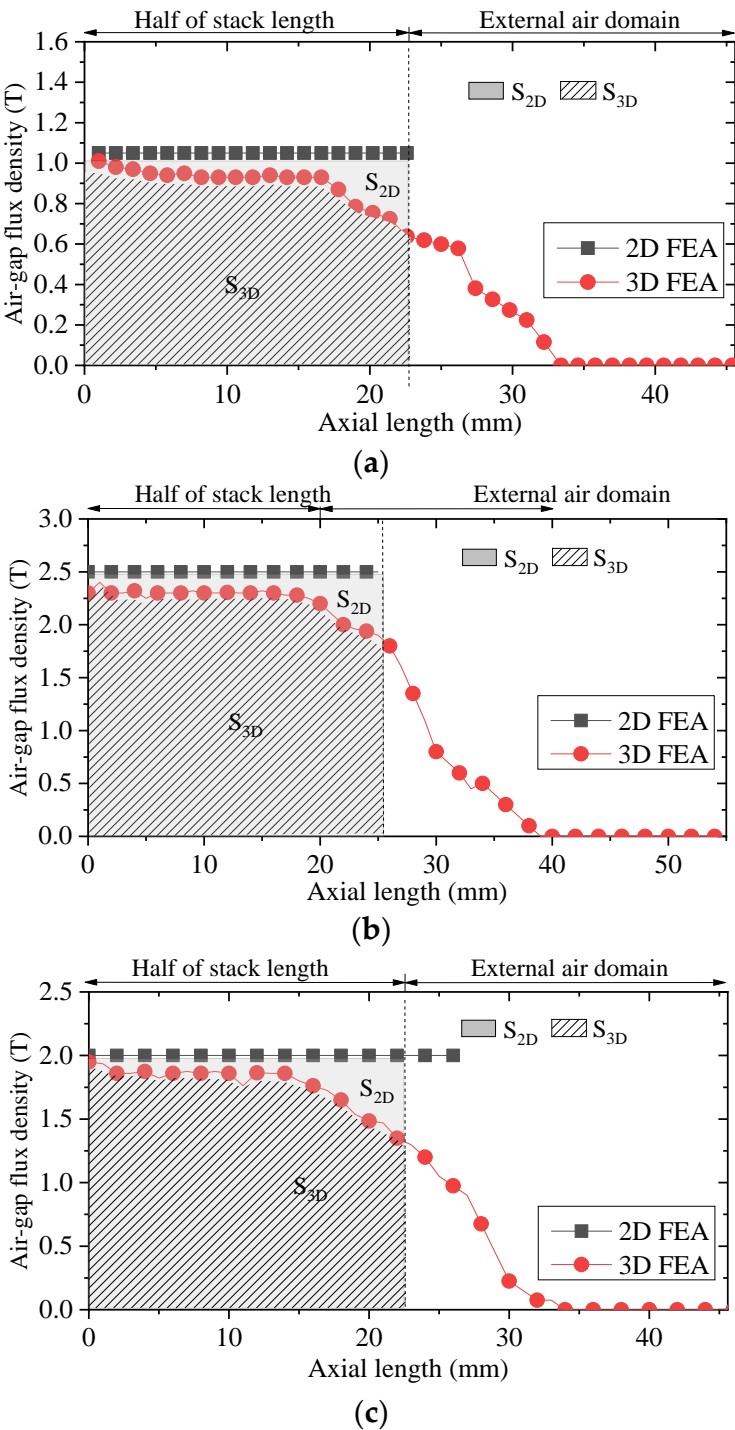

**Figure 11.** Amplitude variations of air-gap PM flux density in axial direction. (**a**) IPM machine, (**b**) SPM-FS machine, (**c**) RPM-FS machine.

### 4.4. Torque Characteristic

Figure 12 compares the output torque of the IPM, SPM- and RPM-FS machines at the rated current of 150 Arms, where the average torque ($T_{ave}$) is 109.5 Nm, 136 Nm, 106.7 Nm and 129 Nm, respectively. Obviously, due to the 27.4% larger volume, the torque of the SPM-FS machine is maximum. For the RPM-FS machine, the torque is larger than the IPM machine by 17.8% with the same volume, which means the torque density is strongest. Moreover, the rated torque ripple of the IPM machine is largest, which is a drawback of the IPM machine.

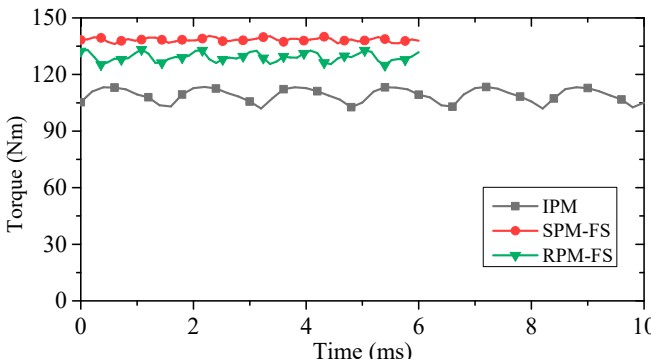

**Figure 12.** Rated torque waveforms of the three machines @150Arms.

Further, to compare the overload capacity of the three machines, Figure 13 shows the average torque values at different phase currents. It can be seen that the RPM-FS machine always exhibits a larger torque than the IPM machine. However, for the SPM-FS machine the torque values are maximum due to the larger volume. The stator outer diameter and silicon steel sheet stacking thickness of the SPM-FS machine are 260 mm and 53 mm, respectively. The stator outer diameter and stacking thickness of the RPM-FS machine and IPM machine are both 253 mm and 45.6 mm, respectively. The PM usage of the SPM-FS machine is 2.2 kg, while the PM usage of the RPM-FS machine and the IPM machine are 0.75 kg and 0.668 kg, respectively. It is precisely due to these two reasons that the maximum output torque of the SPM-FS machine is higher than that of the other two PM machines. The above results verify that the torque capability of the RPM-FS machine can be majorly improved by moving the PMs from stator to rotor, with fewer PMs consumed.

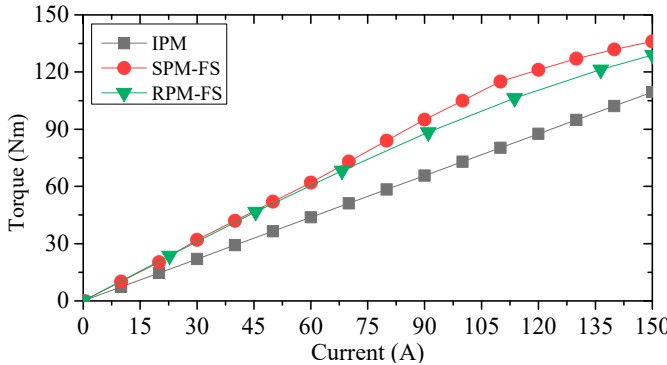

**Figure 13.** Output torque versus phase current of the three machines.

*4.5. Flux-Weakening Ability*

The constant power speed range is a key characteristic of ISG machines for HEV applications. Generally, the flux-weakening ability of PM brushless machines can be expressed by a coefficient $k_{fw}$ [22].

$$k_{fw} = \frac{\Psi_{pm}}{\Psi_{pm} - L_d i_d} \tag{6}$$

Under the condition of $U_{dc}$ = 158 V and $I_{ph}$ = 150 Arms, the torque-speed ($T_e - n$) and power-speed ($P - n$) characteristics of the three machines are simulated via MTPA control in constant torque region and flux weakening control in constant power region [23], and the results are shown in Figure 14. It can be found that the rated speed of 1000 rpm can be achieved by all three machines. The average torque of the RPM-FS machine is larger than that of the IPM machine within the full speed range (0, 6000 rpm). Correspondingly, the output power of the RPM-FS machines is slightly and remarkably larger than that of the

SPM-FS and IPM machines, respectively. Considering the same volume of the RPM-FS and IPM machines, the torque density and power density of the RPM-FS machines are both best and offer promising potential for ISG applications.

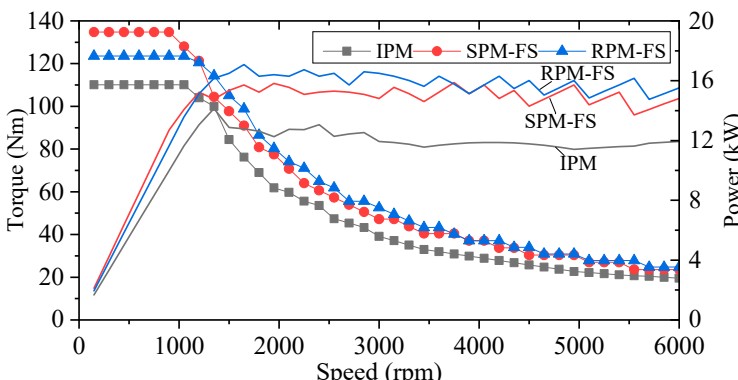

**Figure 14.** Electromagnetic torque and power vs. speed of the three machines @ Udc = 158 V and Iph = 150 Arms.

### 4.6. Thermal Analysis

Due to the various harsh environments and extreme operating conditions faced by vehicle components, good comprehensive temperature management is crucial for extending machine life and preventing demagnetization of PMs. The working points of vehicle components are complex and variable. In order to compare the heat dissipation of the analyzed three machines, a typical working point is selected. The speed and torque of this working point are 1500 rpm and 80 Nm, respectively. The machine is cooled by housing water jackets with the same dimensions, and the coolant is ethylene glycol. The ambient temperature and coolant temperature are both set as 55 °C, and the inlet flow is 18 L/min. The temperature distribution of the three machines is shown in Figure 15. It can be seen that at the same output power conditions, the highest temperature point of all three machines appears at the end winding. The winding temperature of the RPM-FS machine is the lowest, with a value of 133 °C. The winding end temperatures of IPM machine and SPM-FS machine are 145 °C and 144.6 °C, respectively. Due to the higher rotor loss of the RPM-FS machine compared to the IPM machine, the PM of the RPM-FS machine is about 12 °C higher than that of the IPM machine. Due to the PM being located on the stator side of the SPM-FS machine, the heat generated by the PM can be directly transferred to the casing. This is beneficial for reducing the temperature rise.

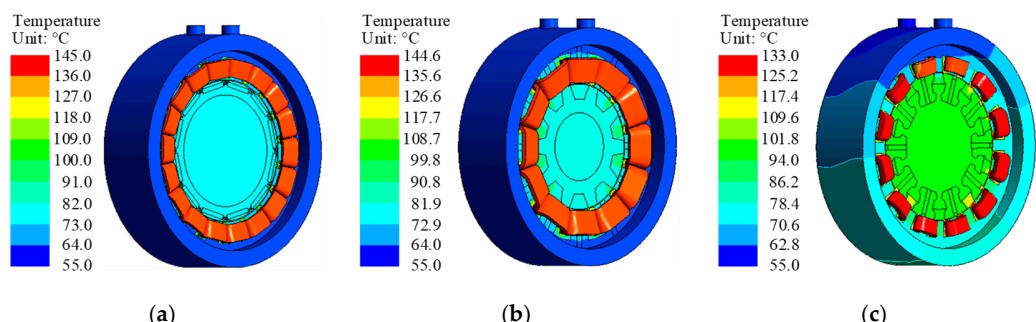

**Figure 15.** Temperature distribution of the three machines. (**a**) IPM machine, (**b**) SPM-FS machine, (**c**) RPM-FS machine.

### 4.7. Stress Deformation Analysis

Since the permanent magnets of the IPM and RPM-FS machines are located in the rotor, and the rotor of the RPM-FS machine is a non-integral structure, the RPM-FS machine is affected by centrifugal force when rotating at high-speed conditions. Ensuring rotor

structural strength is a difficult challenge in design. In order to prevent the occurrence of large deformation and breakage of the rotor when rotating at high-speed conditions, stress deformation analysis of the rotor is required. The peak speed of all three machines is 6000 rpm. Due to the influence of material consistency and processing technology, a safety margin coefficient of 1.2 times is considered in the stress deformation analysis to ensure the reliability of the rotor. Figure 16 shows the stress deformation of the three machines at 7200 rpm. It can be seen that the maximum rotor deformation of the IPM, SPM-FS and RPM-FS machines during operation is 12.27 μm, 1.01 μm and 10.98 μm, respectively. The maximum rotor deformation does not exceed 1/10 of the effective length of the air-gap, and the maximum stress is 111.6 MPa, 9.22 MPa and 83.7 MPa, respectively, which does not exceed the yield strength of the material.

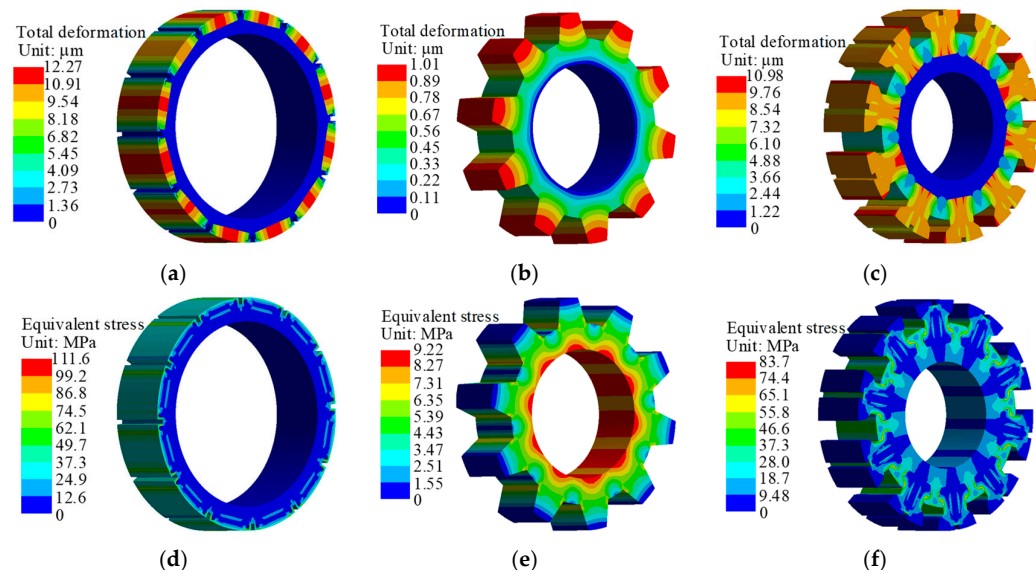

**Figure 16.** Stress and deformation of the three machines. (**a**) IPM machine rotor, (**b**) SPM-FS machine rotor, (**c**) RPM-FS machine rotor, (**d**) IPM machine rotor, (**e**) SPM-FS machine rotor, (**f**) RPM-FS machine rotor.

### 4.8. Comprehensive Efficiency

The efficiency of the three machines is shown in Figure 17. In comparing the comprehensive efficiency of the three machines under drive cyclic conditions, it is assumed that the total weight of the vehicle is 1360 kg, the drag coefficient is 0.26, the windward area is 1.7 $m^2$, the rolling resistance coefficient is 0.0054, the wheel radius is 0.2 m, the transmission efficiency is 90% and the gear ratio is 4.1. The selected cycle condition is China Automotive Test Cycle (CLTC), as shown in Figure 17a, under which the comprehensive efficiency of the three motors is 90.1%, 88.7% and 91.6%, respectively.

### 4.9. Cost Analysis

In the practical application, the cost and performance are equally important. Hence, in addition to the performance discussed above, the material masses and the cost are analyzed. The key components consumption of the three machines are calculated as shown in Table 8, where the masses of the rotor shaft and bearing are not taken into account. It can be found that each component mass of the RPM-FS machine is least, especially for the PM consumption, 10.9% lower than that of the IPM machine. The SPM-FS machine results in the largest effective mass and the PM mass is, respectively 2.93 and 3.29 times that of the IPM and RPM-FS machines. Obviously, the RPM-FS machine is the most advantageous of the three machines and its power per mass ratio and power per volume ratio are both largest.

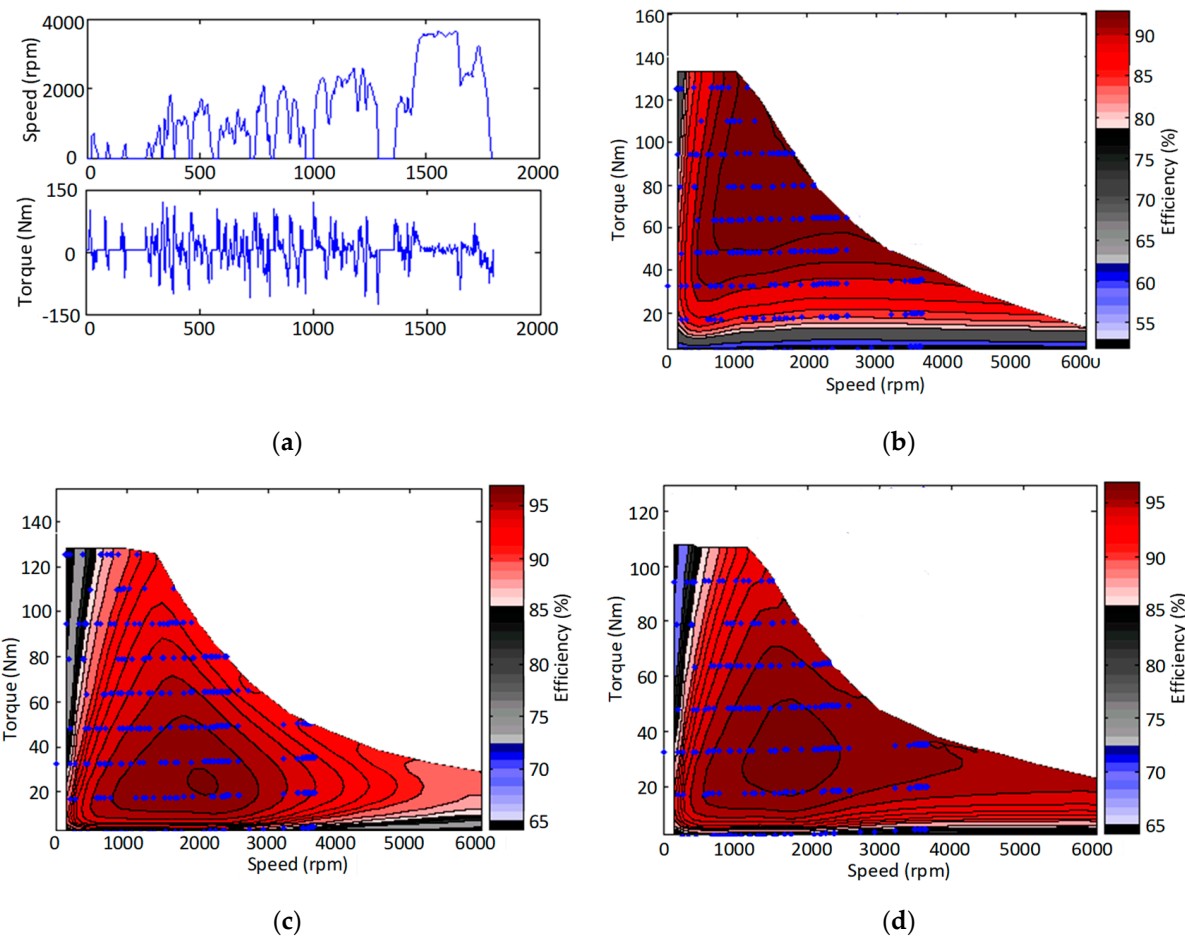

**Figure 17.** Comprehensive efficiency based on cycle conditions. (**a**) CLTC drive cycle, (**b**) SPM-FS machine, (**c**) RPM-FS machine, (**d**) IPM machine.

**Table 8.** Comparison of material masses of the three machines.

| Parameters | IPM | SPM-FS | RPM-FS |
|---|---|---|---|
| Stator lamination mass (kg) | 7.19 | 6.23 | 5.9 |
| Rotor lamination mass (kg) | 2.06 | 4.98 | 2.06 |
| Copper mass (kg) | 2.56 | 2.36 | 1.45 |
| PM mass (kg) | 0.75 | 2.2 | 0.668 |
| Cost of silicon steel sheet (USD) | 13.4 | 16.3 | 11.5 |
| Cost of permanent magnets (USD) | 18.6 | 17.1 | 10.5 |
| Cost of copper (USD) | 21.75 | 63.8 | 19.4 |
| Total effective mass (kg) | 12.6 | 15.77 | 10.07 |
| Maximum power (kW) | 14.1 | 15.8 | 16.7 |

## 5. Experimental Verification

In this section, to validate the FEM simulations, a prototyped SPM-FS machine is built as shown in Figure 18, according to the design parameters listed in Table 1. An experimental platform, shown in Figure 19, is built for testing, including a dynamometer, a DC power supply, a DSP-based digital controller and a transformer. A YOKOGAWA power analyzer and a Tektronix TDS2001C digital oscilloscope are used. The dynamometer machine is driven by an inverter and connected to the prototyped SPM-FS machine by a torque transducer. A resolver is employed to measure the rotor position and speed.

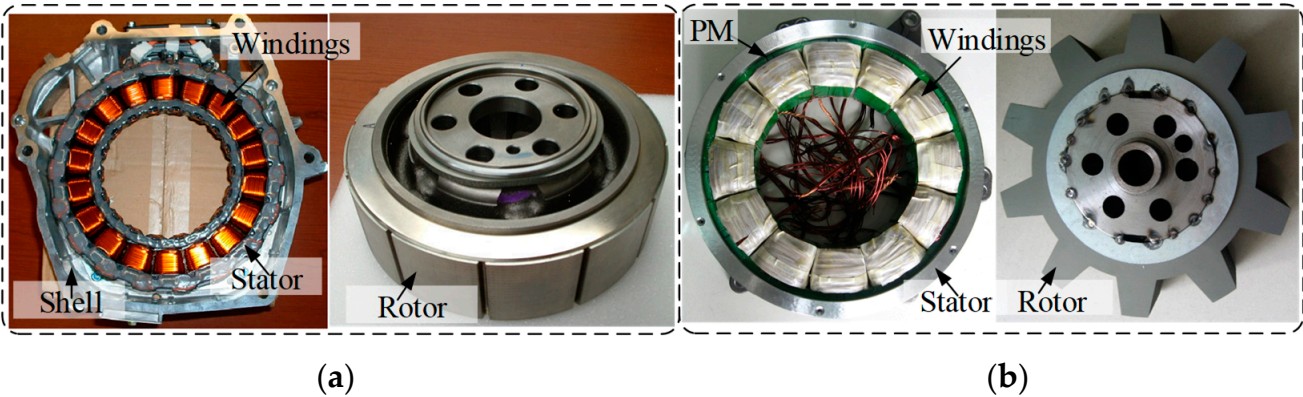

**Figure 18.** The two experimental machines. (**a**) IPM machine [24–26], (**b**) SPM-FS machine.

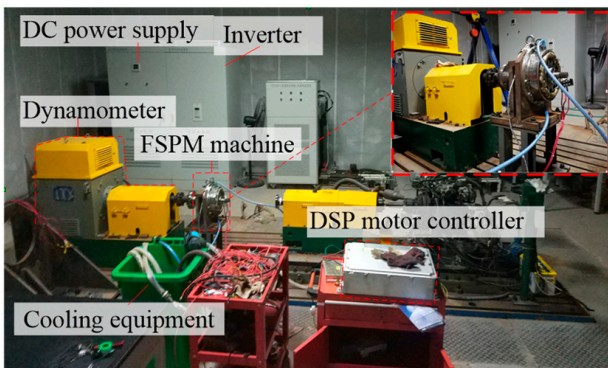

**Figure 19.** Experimental platform setup.

Firstly, the prototyped SPM-FS machine is driven at the rated speed of 1000 pm. The FEM predicted and measured back-EMF waveforms are compared in Figure 20. The static torque versus current angle is compared in Figure 21 with a phase current of 25 Arms. It can be seen that the measured optimal current angle is 10°, which agrees with the predicted result well. Finally, the predicted and experimental average torque values under different phases of the SPM-FS and IPM machines are compared in Figure 22, with satisfied agreements. Overall, agreements are achieved between the FEM predictions and measured results considering manufacturing and testing tolerances.

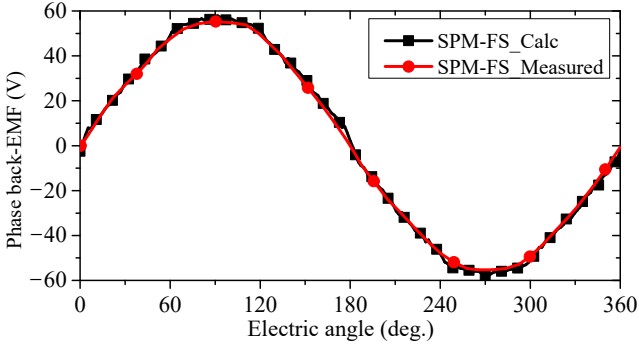

**Figure 20.** Predicted and measured phase back-EMF of the SPM-FS machine @1000 rpm.

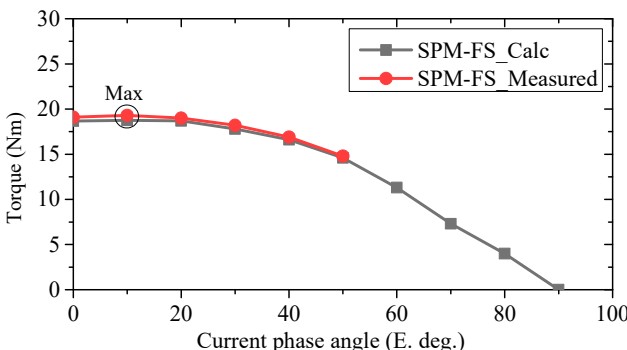

**Figure 21.** Static torque vs. current angle of the SPM-FS machine @ Iph = 25 Arms.

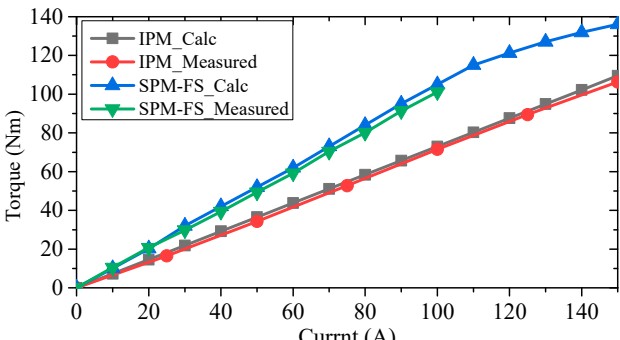

**Figure 22.** Predicted and measured torque-current characteristics of the SPM-FS and IPM machines.

## 6. Conclusions

In this paper, a comprehensive comparison of IPM, SPM- and RPM-FS machines for ISG application is conducted with similar machine dimensions, material properties, and current density, and DC-link bus voltage. The electromagnetic torque performances of the three machines are compared and evaluated. Some conclusions can be summarized as follows.

1. Both the stator- and rotor-PM flux-switching machines exhibit comparable torque capability with that of the IPM machine. However, the SPM-FS machine suffers from limited overload ability due to stator tooth saturation and unfavorable consumption of magnets when the PMs are located in the stator.
2. The RPM-FS machine exhibits a larger torque and overload capability without the space competition pressure from armature windings and PMs. Meanwhile, the amount of PMs has also been greatly reduced, which is favorable for cost. Hence, the RPM-FS machine is a promising candidate for ISG applications.

**Author Contributions:** Conceptualization, W.H.; Methodology, W.Y. and Z.W.; Software, W.Y.; Investigation, W.Y. All authors have read and agreed to the published version of the manuscript.

**Funding:** This research was funded by the National Science Fund for Distinguished Young Scholars under Grant 51825701, the Jiangsu Carbon Peak Carbon Neutralization Science and Technology Innovation Special Fund under Grant BE2022032-1 and the Fundamental Research Funds for the Central Universities under Grant RF1028623013.

**Data Availability Statement:** The data presented in this study are available on request from the corresponding author. The data are not publicly available due to the fact that the project is still in progress.

**Conflicts of Interest:** The authors declare no conflict of interest.

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
