# Peer review of "Performance Evaluation of Stator/Rotor-PM Flux-Switching Machines and Interior Rotor-PM Machine for Hybrid Electric Vehicles"

_wevj, doi:10.3390/wevj14060139_

Round 1

Reviewer 1 Report

The paper presents the investigation by comparing the motor performances for three different permanent magnet (PM) machines, i.e.: (a) the commercial PM machine using interior PM (IPM) produced by Honda that employed in Honda Civic; (b) stator PM flux switching (SPM-FS) machines; and rotor PM flux switching (RPM-FS) machines, where the PM machines aforementioned are used as integrated-starter-generator (ISG) in hybrid electric vehicles (HEVs).  Firstly, the performances of IPM machine and SPM-FS machine are compared, i.e., nol-load flux density, flux linkage, back-EMF, cogging torque, current angle on electromagnetic torque (include E-SPM-FS), and, torque vs current density (include E-SPM-FS). Then, the RPM-FS machine is introduced, and the optimal RPM-FS machine is revealed considering multi-objective problem by using the genetic algorithm (GA) that plugged-in the finite element (FE) software.  The paper reveals the pareto front optimality of RPM-FS machine in the context of two perspectives, such as, output torque, torque ripple, & efficiency, and output torque, efficiency, & material cost, by setting several parameter variables and optimization objectives with its boundaries.  After that, the motor performances of three PM machines (IPM, SPM-FS, & RPM-FS machines) are evaluated and compared, which are: (a) open-circuit analysis (field distribution across the PM machines, radial component of magnetic field distribution in the air-gap with its harmonic distribution); (b) cogging torque; (c) amplitude of air-gap flux density in axial direction; (d) rated output torque; (e) torque vs current; (f) torque & power vs speed; and (g) material masses. A prototype of SPM-FS machine has been built to compare the FEM results, which are: (a) phase back-EMF (only SPM-FS machine); (b) static torque vs current phase angle (only SPM-FS machine); and torque vs current characteristic (SPM-FS machine & IPM machine).  The research works are interesting, however, the manuscript is can be further improved for better comparative study.

Hope the suggestions below can help in improving the content of the manuscript:

- the investigation of thermal study and stress deformation can be included, since the materials are included in the simulation and 3D modelling are included; at the same time, thermal and stress deformation are also the important factors to be considered, if comparing to the commercial PM machine;

- the abbreviations should be defined once before we can use the abbreviation, in each section, such as, abstract & body text, lines 12, 17, 118, 123, 138, etc;

- all variables in the figure should be defined in the text, such as, line 78, Figure 1, and this step should apply in the entire manuscript;

- please elaborate on how to obtain the specification of IPM machine used in Honda Civic, and its geometrical dimension, at the same time, please show the ethical approval on using the commercial IPM machine from Honda;

- redraw Figure 1 (c) for better illustration;

- Figure 1 (c), please elaborate the difference of hm and wpm of IPM and SPM-FS machines;

- Table 1, g0, ksio, should be subscribed, and, please check the format related to this issue in the entire manuscript;

- Table 1, please elaborate the meaning of 34.77 x 5.35 and 39 x 11.65;

- Table 1, please include thickness of PM, wire size of winding, wire arrangement of withstanding 150 Arms;

- line 118, “equs” can be deleted;

- winding arrangement of every PM machines can be included;

- section 2.3.2, please elaborate the reason why there are difference of motor performances among IPM and SPM-FS machines;

- section 2.3.3, comparison in term of using total volume of PM machine in this research that related to the output torque seems on reasonable, since the PMs are playing the important role to provide energy during the power conversion; hence, perhaps, the ratio of PM volume vs total volume of PM machine and/or the ratio of copper winding volume vs total volume of PM machine can give us more insight about this phenomenon;

- section 2.3.3, the E-SPM-FS machine can be excluded from the manuscript, since the analysis of E-SPM-FS machine is not the main objective of the research work;

- line 172, please avoid using “above”;

- Figure 4, the label of x- and y-axes of the small graph should be included;

- please state clearly either electrical degree or mechanical degree in every graph on the axis label, such as, Figure 4;

- Figure 6, please include the specifications and dimensions of RPM-FS machine;

- Figure 6, the RPM-FS machine seems having 24 slot instead of 12 slot, please have a check;

- line 204, please elaborate the GA that has been employed in this research, and why using GA;

- Table 2, the table should be separated into 2 tables, one for performance index and another for optimal variables, the peak cogging torque is repeated in the table;

- Table 2, the equations from stator tooth width arc to rotor tooth top width arc should be defined in text, and further elaborated;

- line 217, (4), please elaborate how to obtain the coefficient of 7.25, 29, and 1.45, as well as the formation of equation (4), please elaborate how to obtain the masses of copper, PMs, stator, and rotor silicon steel sheet;

- Table 3, Table 4, (3), (4), please elaborate how the optimization flow of taking place;

- Figure 7 (a), perhaps torque ripple in percentage is preferable for pareto front consideration;

- Table 5, the table should be separated into 2 tables;

- Figure 7, please elaborate on choosing the optimal RPS-FS machine;

- line 259 & 260, the harmonics numbers should be revised;

- Figure 9, please include the tangential component of magnetic flux density with its harmonic;

- Figure 10, please indicate the meaning of 2 double headed arrow;

- line 279, please indicate mechanical degree or electrical degree;

- line 279, does the statement of “peak cogging torque is inversely proportional to LCM(Pr, Ps)” valid for all PM machines;

- line 295, please elaborate on how to compute the end-part factor;

- Figure 11, please elaborate how to compute the air-gap flux density;

- Figure 12, Figure 17, Figure 19, please redraw the figure for better illustration;

- Line 322, please further elaborate they the SPM-FS machine produces maximum torque as compare with that of the other PM machines;

- Table 6, please elaborate how to obtain the total effective mass;

- section 5, please include the experimental comparison of RPS-FS machine, as proposed by the in the manuscript;

- line 373, please revise the 10o;

- section 5, please include the experimental evaluation and comparison for IPM, SPM-FS, and RPM-FS machines with the FEA predicted results, which include: (a) phase back-EMF; (b) cogging torque; (c) static torque vs current angle; (d) electromagnetic torque; (d) torque-current characteristic; (e) torque-speed curve; and (f) efficiency.

A little grammatically error can be improved.

Author Response

To Reviewers:

Re: No. wevj-2390139: Performance Evaluation of Stator/Rotor-PM Flux-Switching Machines and Interior Rotor-PM Machine for Hybrid Electric Vehicles.

My co-authors and I would like to thank the reviewers and the editor for their constructive and helpful comments and suggestions. All of their comments and suggestions have been taken into full consideration in this revised version.

All of the modifications are highlighted in RED in the paper. The detailed responses to reviewers’ comments are enclosed.

Yours Sincerely,

Dr. Zhongze Wu

School of Electrical Engineering, Southeast University

No. 2 Si-Pai Lou, Nanjing, China

Reviewer 2 Report

In this paper, the existing motor and two designed motors were compared.

It seems necessary to add the following.

1. HV should be designed as an operating cycle rather than as a simple load point. It is necessary to add how the efficiency are when compared with the operation cycle.

2. The material costs should be added in Table 6.

Author Response

(The authors gave the same response as above.)
